# Effects of Current Annealing on Thermal Conductivity of Carbon Nanotubes

**DOI:** 10.3390/nano12010083

**Published:** 2021-12-29

**Authors:** Huan Lin, Jinbo Xu, Fuhua Shen, Lijun Zhang, Shen Xu, Hua Dong, Siyi Luo

**Affiliations:** 1School of Environmental and Municipal Engineering, Qingdao University of Technology, Qingdao 266033, China; xujinbo0214@163.com (J.X.); Shenfh0126@163.com (F.S.); LJunZhang@outlook.com (L.Z.); dhua1959@hotmail.com (H.D.); luosiyi666@126.com (S.L.); 2School of Mechanical and Automotive Engineering, Shanghai University of Engineering Science, Shanghai 201620, China; shxu16@sues.edu.cn

**Keywords:** carbon nanotubes, current annealing, thermal conductivity, graphitization

## Abstract

This work documents the annealing effect on the thermal conductivity of nanotube film (CNTB) and carbon nanotube fiber (CNTF). The thermal properties of carbon nanotube samples are measured by using the transient electro-thermal (TET) technique, and the experimental phenomena are analyzed based on numerical simulation. During the current annealing treatment, CNTB1 always maintains the negative temperature coefficient of resistance (TCR), and its thermal diffusivity increases gradually. When the annealing current is 200 mA, it increases by 33.62%. However, with the increase of annealing current, the TCR of CNTB2 changes from positive to negative. The disparity between CNTB2 and CNTB1 suggests that they have different physical properties and even structures along their lengths. The high-level thermal diffusivity of CNTB2 and CNTF are 2.28–2.46 times and 1.65–3.85 times higher than the lower one. The results show that the decrease of the thermal diffusivity for CNTB2 and CNTF is mainly caused by enhanced Umklapp scattering, the high thermal resistance and torsional sliding during high temperature heating.

## 1. Introduction

Carbon nanotubes (CNTs) have progressively attracted researchers’ attention for their lightweight, small size and acceptable flexibility. Moreover, they have both excellent electrical and thermal conductivity, which determines that they have application value and development potential as a high-performance reinforcement material [1]. However, in practical applications, they have failed to reflect the excellent properties of CNTs because they are dispersed randomly and prone to agglomeration [2]. Previous studies on CNTs have concentrated on the preparation [3,4], the optimization of the mechanical properties of CNTs composites [5,6] and the enhancement of the effectiveness of CNTs as capacitor electrode materials [7]. In recent years, studies have shown that high temperature annealing is an effective approach to improve the structure and thermal conductivity of carbon materials [8,9]. Chen et al. [10] found that thermal treatment is capable of repairing the structure of graphene and making graphene sheets accumulate more regularly. Thereby, the thermal conductivity is increased. Mayhew et al. [11] found that the thermal conductivity of carbon nanofibers could be increased by nearly 40 times after annealing at 2800 °C for 20 h. However, the effect of high temperature annealing on the thermal conductivity of carbon materials is complicated, for which a mechanism has yet to be figured out.

The traditional method of annealing at present is indirect annealing in the furnace, and the current annealing in this experiment has the superiorities of straightforward and efficient in situ measurement compared with the traditional annealing. Current annealing can be accomplished in a few seconds for the sample annealing treatment. The sample is always in the invariable equipment before annealing until the sample is burned down after complete annealing to measure thermal conductivity, which reduces the measurement errors caused by factors such as the pollution of the area under measurement and the different qualities of the sample.

In this work, the focus is on the current annealing effect on CNTB and CNTF. The data is collected by using the TET technique, and the variation of thermal diffusivity with annealing current is analyzed. The evolution of material microstructures is studied based on the variation of their thermal properties.

## 2. Materials and Methods

### 2.1. Experimental Materials

The CNTB (Suzhou Tanfeng Graphene Technology Co., Ltd., Suzhou, China) was prepared from multi-walled carbon nanotubes (MWCNTs) by floating catalysis to pattern a film with the size of 10 cm × 10 cm (Figure 1a), with an electrical conductivity of (0.8–3) × 10^−5^ S/m, a strength of 60–100 MPa and a density of 400 kg/m^3^. Figure 1b, the SEM image of CNTB, clearly shows that the film is made of innumerable carbon nanotubes arranged desultorily. Moreover, the carbon nanotubes have non-uniform diameters and randomly overlap with others, which makes the surface of the prepared film rough to a certain extent. The tested samples were prepared along the length direction of the square CNTB film (CNTB1) and along the other side of the CNTB film (CNTB2), as shown in Figure 1a.

The carbon nanotube fiber (CNTF, Nanjing JCNANO Tech Co., Ltd., Nanjing, China) was also prepared from MWCNTs by using the floating catalyst method, with a density of 675.43 kg/m^3^, the strength of 310–500 MPa, the modulus of 10 GPa and a tensile rate of 20–30%. From its SEM images (Figure 1c,d), the CNTF sample had a certain degree of twisted texture and roughness, and there is a cross-overlap as well.

### 2.2. Transient Electro-Thermal Technique

The thermal conductivities of carbon nanotube materials were measured by using TET. Figure 2 shows the schematic diagram of the TET experimental system. The samples were suspended on a sample holder with two separate electrodes. A small amount of silver paste was applied to stabilize both ends of the samples on the electrodes to reduce the contact resistance between the samples and electrodes. Then the sample holder was placed into a vacuum chamber to exclude heat convection. As the air pressure in the vacuum chamber during the measurement was maintained at 1 × 10^−3^ mbar, the heat convection effect on the measured thermal diffusivity was negligible. A step current supplied by a current source (KEITHLEY 2611A, Keithley Instruments Inc., Cleveland, OH, USA) was fed to the samples to generate Joule heating in the samples. The thermal diffusivity of the samples was obtained by analyzing the temperature rise curve with a physical model discussed below.

As the CNTB and CNTF samples have a large aspect ratio, a one-dimensional heat transfer model was used for analysis [12]. The average temperature change of the sample directly affects the variation of the voltage over the sample as [13]:(1)$$V_{s}{}_{\mathrm{ample}}=IR_{0}+I\eta \frac{4q_{0}L^{2}}{k\pi^{2}}\sum_{m=1}^{\infty }\frac{1-\exp[(-{\left(2m-1\right)}^{2}\pi^{2}\alpha_{\mathrm{eff}}t)/L^{2}]}{{\left(2m-1\right)}^{4}}$$
where *η* is the temperature coefficient of resistance, *q*_0_ is the electrical heating power per unit of volume, *k* is the thermal conductivity, *α*_eff_ is the effective thermal diffusivity of the sample.

Defining a normalized average temperature rise as $T^{*}=(T_{t}-T_{0})/(T_{t}{}_{\to \infty }-T_{0})$, it can be expressed as [13,14]:(2)$$T^{*}=\frac{48}{\pi^{4}}\prod_{m=1}^{\infty }\frac{1-{\left(-1\right)}^{m}}{m^{2}}\frac{1-\exp[-m^{2}\pi^{2}\alpha_{\mathrm{eff}}t/L^{2}]\;}{m^{2}}$$

When a step current is applied to a sample, its resistance varies with the temperature, which leads to a change of voltage. Therefore, the experimental value of the normalized temperature rise $T^{*}{}_{\exp}$ can be calculated by the change of voltage as: $T^{*}{}_{\exp}=V^{*}=(V_{s}{}_{\mathrm{ample}}-V_{0})/(V_{\infty }-V_{0})$, where $V_{0}$ and $V_{\infty }$ are the initial and steady-state voltages across the sample, respectively. The *V-t* curve of the sample was recorded by an oscilloscope (DSO-X3052A, Agilent Technologies Inc, Santa Clara, CA, USA); combined with Equation (1), the thermal diffusivity (α) of the sample was obtained, which was used to fit the normalized temperature curve. The α value with the best fitting was determined as *α*_eff_, and associates with the density (ρ) and specific heat capacity ($c_{p}$) of the sample, its effective thermal conductivity (*k*_eff_) can be obtained as calculated by: $k_{\mathrm{eff}}=\rho c_{p}\alpha_{\mathrm{eff}}$.

### 2.3. Experimental Procedure

The current annealing was applied to the sample in the same experimental setup as TET in this study. After the sample was laid in a vacuum chamber, the *α*_eff_ of the sample at room temperature was first measured using a step current with low intensity in order to raise the temperature as small as possible. Then a Direct Current (DC) with high intensity was applied to the same sample to generate large heat in the sample and complete the thermal annealing. The thermal annealing lasted for more than 120 s to ensure thermal equilibrium for one run. The second *α*_eff_ was measured after the sample finished the first annealing run. The annealing run and in-situ TET measurement were then performed alternatively by switching the form of the current between the large DC current (for annealing) and small step current (for TET measurement). The DC current increases a little at a time until the sample is burned down. This method can fully realize the effective monitoring of the structural variation in the annealing process. The error of transferring samples between the different annealing and measurement devices is successfully avoided. This is the whole current annealing procedure, in which the effects of current annealing on thermal conductivity are investigated.

## 3. Results and Discussions

### 3.1. Results

#### 3.1.1. Positive Effect of Annealing on CNTB1

The annealing process and in-situ TET measurement are first applied to CNTB1 (length: 3620.71 μm; width: 288.42 μm; thickness: 25 μm). The decreasing voltage along with time demonstrate that the MWCNTs dominate the heat conduction in CNTB1 because most carbon materials have a negative temperature coefficient of resistance (TCR) [15]. The varying trend is fitted by Equation (2), and the best-fitted *α* is the effective thermal diffusivity *α*_eff_ of CNTB1. The detailed annealing current *I*_a_ and *α*_eff_ of CNTB1 are summarized in Table 1. With the increase in the annealing current, *α*_eff_ of CNTB1 increases likewise. When less heat is added, *α*_eff_ changes little. As the sample is heated in a vacuum chamber, the thermal reduction would occur in the sample to remove impurity induced in the sample production. With the increase of the current, the degree of reduction is enhanced, and the microstructure is optimized. Thus, the thermal conductivity and *α*_eff_ of CNTB1 rise. The annealing current of burning the sample is 210 mA, and when the annealing current is 200 mA, the thermal diffusivity increases by 33.62%.

#### 3.1.2. Observation of Thermal Diffusivity Jump on CNTB2 and CNTF

In the process of current annealing experiments for CNTB2 (length: 3657.23 μm; width: 260.23 μm; thickness: 25 μm) and CNTF (length: 3429.35 μm; diameter: 135 μm), an unusual phenomenon appears where the thermal diffusivity of the sample has a “sudden jump”. Figure 3a shows the voltage evolution of CNTB2 before annealing. It is obvious that the varying trend is different from that of CNTB1, though both of them are cut from the same carbon nanotube conductive film. The disparate phenomenon of CNTB2 and CNTB1 indicates that they have different physical properties and even structures along their length directions, which lead to the occurrence of different heat conduction mechanisms.

Figure 3b–f shows the *V-t* experimental data and the theoretical fitting curves of CNTB2 under different currents: 140 mA, 160 mA, 165 mA, 170 mA and 300 mA. It presents an interesting evolution of the voltage variations. When the annealing current varies from 0 to 140 mA, the voltage of CNTB2 increases gradually under Joule heating and then reaches the steady-state, which is a typical voltage evolution with a positive TCR. However, when the annealing current is above 150 mA, the TET signal for CNTB2 shows a small decrease at the beginning of Joule heating, and then it rises again until reaching the steady-state. With the increase in the annealing current, the descending part becomes noticeable and begins to dominate the TET signal. When the current passes 170 mA, the signal only has a decreasing trend until reaching the steady-state, and the rising portion completely disappears. The sample initiates to perform the normal signal which the carbon materials originally have when the sample has a negative TCR. The “sudden jump” of thermal diffusivity indicates that with the increase of the annealing current, the TCR of CNTB2 and CNTF turns from positive to negative.

CNTF has a similar behavior as CNTB2. Both samples are annealed with a higher and higher current until the sample is burned down. The thermal diffusivities of CNTB2 and CNTF against the annealing currents are shown in Figure 4a,b, respectively. The detailed annealing condition and determined thermal diffusivity are listed in Table 2. The measured thermal diffusivity is divided into two separate data groups. The lower thermal diffusivity group is denoted as *α*_1_, and the higher thermal diffusivity group is denoted as *α*_2_. The corresponding two states before and after switch-on are named State 1 and State 2, respectively. Combining Table 2 and Figure 4, it can be observed that when the annealing currents of CNTB2 and CNTF are 160 mA and 350 mA, the peak thermal diffusivity is 32.01 × 10^−5^ m^2^/s and 19.63 × 10^−5^ m^2^/s. *α*_1_ is much lower than *α*_2_. Wang et al. [16] found that high temperature induces C atoms to act in a thermal flutter in a large range at the structural equilibrium position, and this deformation increases the atomic energy in the local region of MWCNTs. When the threshold of the barrier constraint value is reached, the MWCNTs structure will produce irreversible deformation (the minimum distance between C atoms) and even collapse [17]. It can be further determined that the thermal diffusivity peaks in CNTB2 and CNTF occur at the threshold where instability is produced in MWCNTs’ structures.

#### 3.1.3. Transient Annealing Behavior

The transient voltage variations of CNTB1, CNTB2 and CNTF under different annealing currents are shown in Figure 5. During the process of annealing, the time required by State 1 for CNTB and CNTF decreases. With the increase of the annealing current, State 2 presents complex and diverse changes, and when the annealing current approaches the maximum current that the sample can withstand, the voltage fluctuates greatly. It is clear in Figure 5 that transient voltage variations of CNTB and CNTF appear in a rising trend, suggesting that in the process of annealing, the resistance of the sample increases constantly with the heating temperature rising. This phenomenon is in conflict with the theory that carbon materials have negative TCR in themselves.

As shown in Figure 6a, the normalized resistance (*R*^*^) of both samples has an upward trend with the rise of annealing power (*P*). After several times of annealing, the samples are burnt due to high-temperature heating eventually, and the final breakpoints of samples are shown in Figure 6b–d. CNTB1 is fractured at about 3/4 of the sample length, and CNTB2 and CNTF are both fractured at about 1/2 of their length, which indicates that during the annealing process, the temperature distribution along the length of the sample is non-uniform. The temperature at the breakpoint is the highest, and thus the fracture occurs first.

#### 3.1.4. Temperature Distribution and Thermal Conductivity Change of CNTB

The temperature at different positions along the length of the sample is calculated by using a numerical method, and the results are shown in Figure 7a. The temperature distribution is non-uniform along the length of CNTB1 and CNTB2. The temperature at the middle point is the highest, and those further away from the middle point are lower in temperature. Moreover, the highest temperature of CNTB1 is lower than that of CNTB2, indicating that the thermal conductivity of CNTB1 is smaller than that of CNTB2, which is consistent with the experimental results that the thermal diffusivity of CNTB1 is smaller than that of CNTB2. In addition, the breakpoint of CNTB1 is further away from the middle point, and the inhomogeneity of temperature distribution will also cause the simulation temperature of CNTB1 to be at a low level.

Figure 7b indicates that the Joule heating generated by the current provides the high temperature environment required by the annealing process for CNTB. With the increase of heating energy, the average temperature of samples continues to rise, which makes the annealing process proceed in an orderly manner. The rising rate of the average temperature shows a trend of gradual decrease, which reveals that in the process of annealing, the graphitization levels increase. Nevertheless, the process of graphitization is slowing down, and the degree of graphitization is decreasing, indicating that the graphitizing level is close to the maximum degree of the sample.

Figure 7c shows the variation curves of the thermal conductivity of the middle point (*k*_m_) with the annealing temperature of the middle point (*T*_mid_) of CNTB1 and CNTB2, respectively. *k*_m_ of CNTB2 also appears to make a “sudden jump”, as mentioned above. *k*_m,1_ and *k*_m,2_ correspond to the low thermal conductivity before switch-on and the high thermal conductivity after switch-on of CNTB2. As shown in this figure, with the increase of *T*_mid_, the *k*_m_ of CNTB1 increases, while the *k*_m_ of CNTB2 decreases, which reveals that a single annealing treatment does not always have a positive effect on the thermal conductivity of carbon materials, and the analysis from the overall samples also shows this. *α*_eff_ and the average temperature (*T*_ave_) reflect the average properties of the material. As shown in Figure 7d, with the increase of *T*_ave_, *α*_eff_ of CNTB1 increases, the low *α*_eff,1_ and the high *α*_eff,2_ of CNTB2 decreases.

#### 3.1.5. Temperature Distribution and Thermal Conductivity Change of CNTF

Unlike the “sudden jump” of CNTB2, CNTF does not have low-level thermal conductivity before switch-on. Therefore, the focus of CNTF is only the part with the high thermal conductivity after switch-on. As shown in Figure 8a, the temperature distribution along the length of CNTF is also non-uniform, and the temperature at the middle point is the highest.

Figure 8b shows the variation of the high *k*_m_ with *T*_mid_ after the switch-on of CNTF, indicating that *k*_m_ decreases continuously with the increase of *T*_mid_. Figure 8c is the curve of *T*_ave_ under different heating conditions, and Figure 8d is the variation of *α*_eff_ against *T*_ave_, which shows that the current annealing in this experiment has a certain degree of “negative” effect on the thermal conductivity of CNTF.

### 3.2. Discussions

#### 3.2.1. Mechanism of Phonon Scattering

Heat conduction is normally dominated by phonons of carbon materials [18], whose heat transport is critical to the heat transport of materials. It is speculated that during the current annealing process of CNTB1, the size of graphite microcrystals increases, some of the impurity atoms are moved, and the order of the graphite microcrystalline structure is improved. These transformations reduce the probability of phonons colliding with other phonons, grain boundaries, impurities and edge boundaries in the process of heat transport, thus reducing the amount of phonon scattering and the scattering intensity. The integrity of phonon heat transport improves, and the phonon mean free path increases. Therefore, the thermal conductivity of CNTB1 increases.

The phonon scattering of CNTB2 and CNTF is determined by Umklapp scattering, normal scattering, impurity scattering, phonon-boundary scattering, phonon-defect scattering and thermal contact resistance. Normal scattering generally occurs at low temperatures, and Umklapp scattering is more likely to occur at higher temperatures. Combined with the previous analysis of the “sudden jump”, in State 1, because of the interaction between adjacent pure carbon nanotube (P-CNT) and impurities-embedded carbon nanotube (I-CNT), the thermal diffusivity of material is restrained. Thermal expansion mismatch occurs between P-CNT and I-CNT due to thermal stress, which results in different degrees of stretching or contraction of carbon nanotubes. However, P-CNT is not separated from the surrounding I-CNT, and the restraint of boundary scattering on phonon heat transport is amplified to some extent, resulting in the thermal diffusivity decrease.

In the switch-on state, with the enhancement of annealing, the temperature increases gradually, and the elastic vibration of the lattice increases. Furthermore, more high-frequency phonons are excited, and the number of phonon groups increases, which promotes the heat transport of phonons. The thermal diffusivity of the material is ultimately improved. Because I-CNT is purified and the structure of CNT is optimized continuously, the effect of impurity and boundary scattering on heat conduction is weakened. Therefore, the thermal diffusivity peaks of CNTB2 and CNTF appear at this state.

In State 2, P-CNT and I-CNT are completely separated, and Umklapp scattering is more prevalent. Umklapp scattering is dominant in the phonon scattering mechanism [19], and the process of Umklapp scattering generally causes thermal resistance, which also affects the phonon mean free path. As the annealing temperature increases, the influence of thermal stress goes up. MWCNTs are continuously stretched or compressed, and the structure deforms to different degrees, which is irreversible due to excessive thermal stress [16]. The scattering produced by structural changes is classified as structure scattering. Because of the high thermal stress, the structure of MWCNTs collapses, and the structure scattering is enhanced, which impairs the original larger thermal conductivity of carbon nanotubes. In this state, MWCNT impurities are further removed, and the density of the defect is reduced, which results in the impurity scattering and defect scattering effects being weakened. The high temperature excites more high-frequency phonons to promote heat transport, but it also excites more scattering between phonons, which enhances Umklapp scattering and increases the thermal resistance. Combining with the above factors, the thermal diffusivity of CNTB2 and CNTF decreases.

#### 3.2.2. Effects on Thermal Conductivity

MWCNT torsional sliding induced by high temperature is one of the reasons for the decrease of thermal conductivity. Wang et al. [20] showed that the external load causes the global buckling of carbon nanotubes and even the large torsion. Charlier et al. [21] found that Buckyball molecules rotate freely when it reaches the transition temperature (258 K), which can easily slide into or out of each other, and such movements are unhindered at room temperature. With the increase of annealing, the graphitization degree increases, the structure becomes ordered, and the density of the defect decreases. Chen [22] showed that structure defects limit rotation and sliding between MWCNTs layers. Therefore, the reduction of defect density promotes the rotational migration of CNT to some extent. With the increase of the annealing current, the effect of annealing on the structure gradually increases. When the current increases to a certain degree, the annealing will produce large thermal stress. The stress is equivalent to applying an external load on the material; the CNT rotates under the torque generated by the high temperature, resulting in the deviation of the MWCNTs distribution direction. The axial direction of the MWCNTs cylinder forms a certain angle with the length direction, and even the phenomenon that the MWCNTs is perpendicular to the axial direction appears. The more orderly structure also indicates that more CNT may rotate together along a certain angle and form an ordered arrangement with the axial direction [23]. Therefore, the thermal conductivity of CNTB2 and CNTF decreases.

On the other hand, high annealing temperature may result in greater thermal resistance between MWCNTs, which affects the thermal transport of materials. Gong [24] proved that 1800 ℃ is the appropriate annealing temperature to remove the impurities of MWCNTs. When the annealing temperature rises to 2100 °C, agglomeration occurs in MWCNTs, and aggregates are generated on the surface of the sample. This indicates that with the increase of annealing, the structure of MWCNTs deforms, becomes rough and even agglomerates, so the thermal resistance is increased, and the heat transport is inhibited. Feng et al. [25] showed that high temperature stimulates more phonon scattering, resulting in increased thermal resistance and reduced thermal conductivity. Kim et al. [26] reported that lattice defects of carbon tubes at high temperatures would motivate the Umklapp scattering, causing an increase in thermal resistance. Liu et al. [27] showed that in the graphitization process of carbon fiber, although the crystal size of material increases, the probability of the occurrence of maximum defects and large holes also increased. Therefore, it is speculated that during the annealing process, with the increasing of annealing temperature, the presence of thermal stress leads MWCNT to agglomerate. In the process of graphitization, larger vacancy and hole defects appear in the material, which causes greater thermal resistance, and hinders and inhibits the effective heat transport of phonons. At the beginning of annealing, the TCR of CNTB2 is positive, which means thermal resistance plays an important role in heat transfer. This also explains that the thermal conductivity decreases with the increase of annealing current.

#### 3.2.3. Effects of Annealing on Structure

The Raman characterization of CNTB after burning off is shown in Figure 9. The positions of the measuring point are numbered as 1 to 9, which are closer and closer to the breakpoint from bottom to top. It can be seen that the effect of annealing is ununiformed along the length during the current annealing process, which appears as the effect of annealing is continuously enhanced from both ends to the breakpoint of the sample. Three prominent peaks are at around 1315 cm^−1^, 1586 cm^−1^ and 2610 cm^−1^, which correspond with D peak, G peak and 2G peak, respectively. Figure 10 shows the ratio of peak intensities *I*_D_/*I*_G_ of different points of CNTB. From the endpoint to the breakpoint, the decreased *I*_D_/*I*_G_ ratio means a lower defect. With the enhancement of annealing, the sample becomes orderly, the disorder of internal structure decreases and the defect density also decreases. Moreover, the closer to the breakpoint, the shape of the G peak and 2D peak in Raman become steeper with a narrower line width and stronger intensity, which also indicates the larger crystalline size and structure of the sample is improved gradually.

## 4. Conclusions

This work shows the current annealing effect on the thermal conductivity of CNTB and CNTF. In the annealing process, the thermal diffusivity of CNTB1 increases gradually, and the highest thermal conductivity is 1.34 times the original thermal conductivity. Although both CNTB1 and CNTB2 are cut from the same carbon nanotube conductive film, the thermal diffusivity of CNTB2 has a “sudden jump”, and the TCR of CNTB2 changes from positive to negative. The disparate phenomenon of CNTB2 and CNTB1 indicates that they have different physical properties and even structures along their length directions which lead to the occurrence of different heat conduction mechanisms. The results show that the high-level thermal diffusivity of CNTB2 is 2.28–2.46 times higher than the low one.

The high-level thermal diffusivity of CNTF is 1.65–3.85 times higher than the low one. The main reasons for the decrease of the thermal conductivity of CNTB2 and CNTF are as follows: enhanced Umklapp scattering; torsional sliding occurs in MWCNTs induced by high temperature; high thermal resistance between MWCNTs.

## Figures and Tables

**Figure 1 nanomaterials-12-00083-f001:**
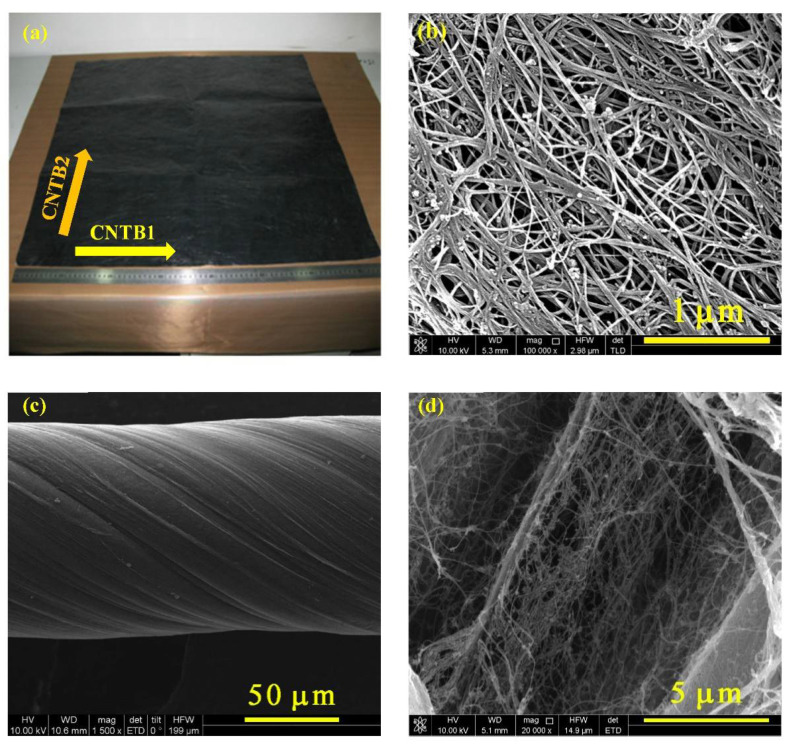
(**a**) The intact carbon nanotube film (CNTB) before the TET experiments. (**b**) The SEM image of the CNTB. The SEM images of CNTF under different magnifications of (**c**) 1500× and (**d**) 20,000×. These indicate CNTB and CNTF are made of innumerable carbon nanotube fibers arranged desultorily.

**Figure 2 nanomaterials-12-00083-f002:**
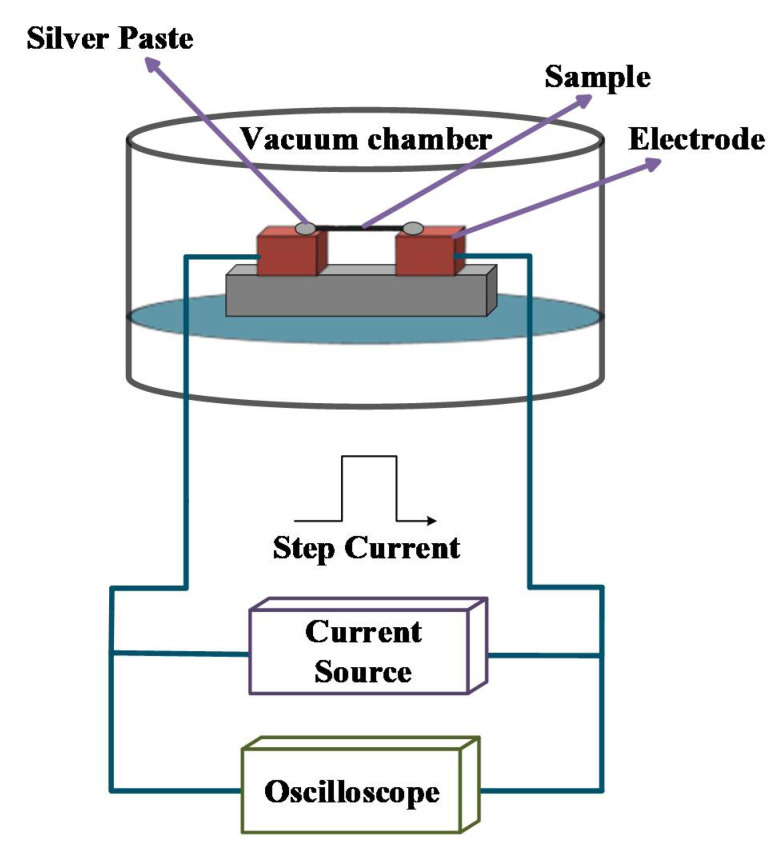
The schematic diagram of the TET experiment. TET is composed of a vacuum chamber, current source and oscilloscope. In the vacuum chamber, the air pressure is less than 1 × 10^−3^ mbar during the measurement. The sample is placed on the electrodes, and a silver paste is applied to the ends for reducing contact resistance.

**Figure 3 nanomaterials-12-00083-f003:**
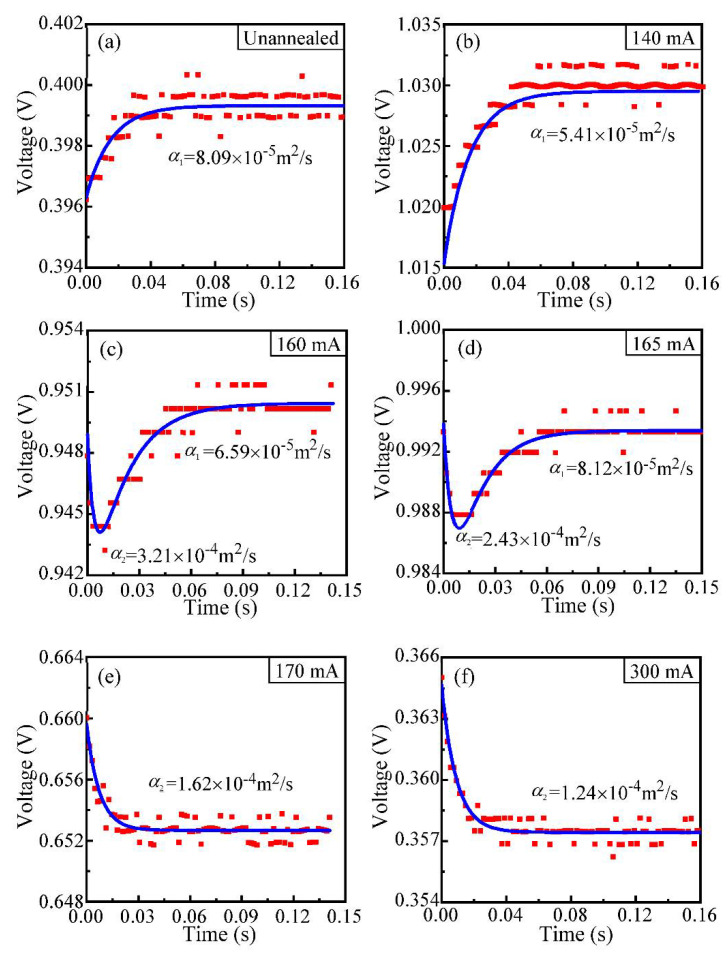
(**a**–**f**) Comparisons between the theoretical fitting and experimental data of CNTB2 for the voltage under different annealing currents. The red squares are experimental data, and the blue lines are the theoretical fitting. (**a**) Corresponding to the unannealed state and the variation of voltage is monotone increasing, which trend is similar to (**b**) under 140 mA. (**c**,**d**) are under 160 mA and 165 mA, respectively. The variations perform as decreasing firstly and then rising to the steady-state. (**e**,**f**) 170 mA and 300 mA, respectively, and the variations are monotone decreasing.

**Figure 4 nanomaterials-12-00083-f004:**
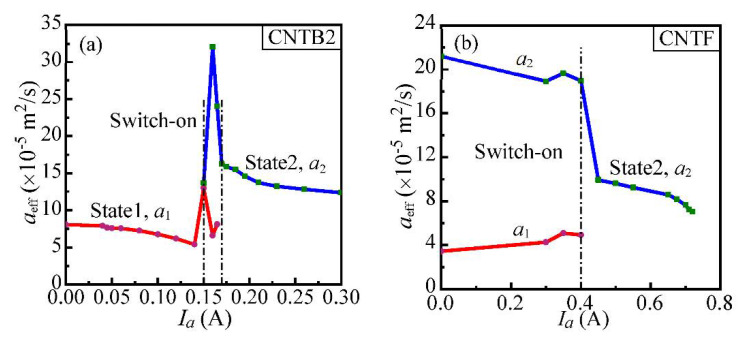
The curve of thermal diffusivity with an annealing current (*I*_a_): (**a**) CNTB2; (**b**) CNTF.

**Figure 5 nanomaterials-12-00083-f005:**
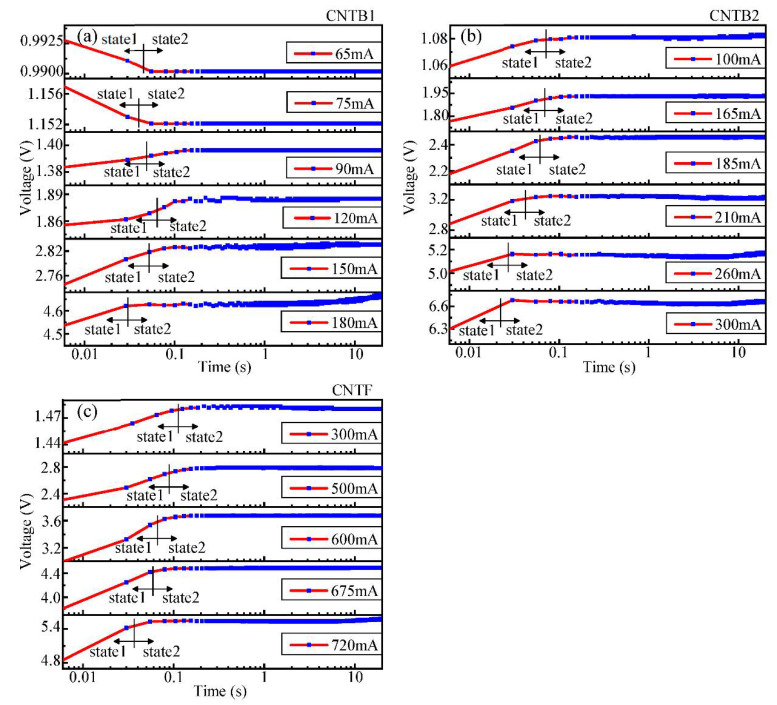
The transient voltage variations with time during annealing: (**a**) CNTB1; (**b**) CNTB2; (**c**) CNTF. The arrows indicate State 1 and State 2. During the current annealing, the time required for each sample in State 1 is reduced.

**Figure 6 nanomaterials-12-00083-f006:**
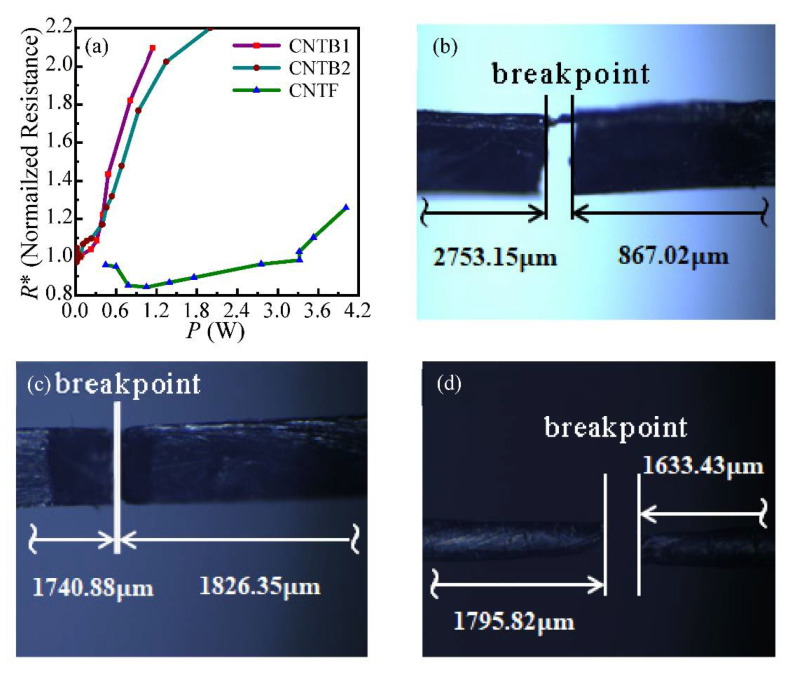
(**a**) Curves of sample-normalized resistance with annealing power. The *R*^*^ of samples has different degrees of increase with the increase of *P*. Diagram of burning down after annealing: (**b**) CNTB1; (**c**) CNTB2; (**d**) CNTF. The locations of breakpoints are shown in the figures.

**Figure 7 nanomaterials-12-00083-f007:**
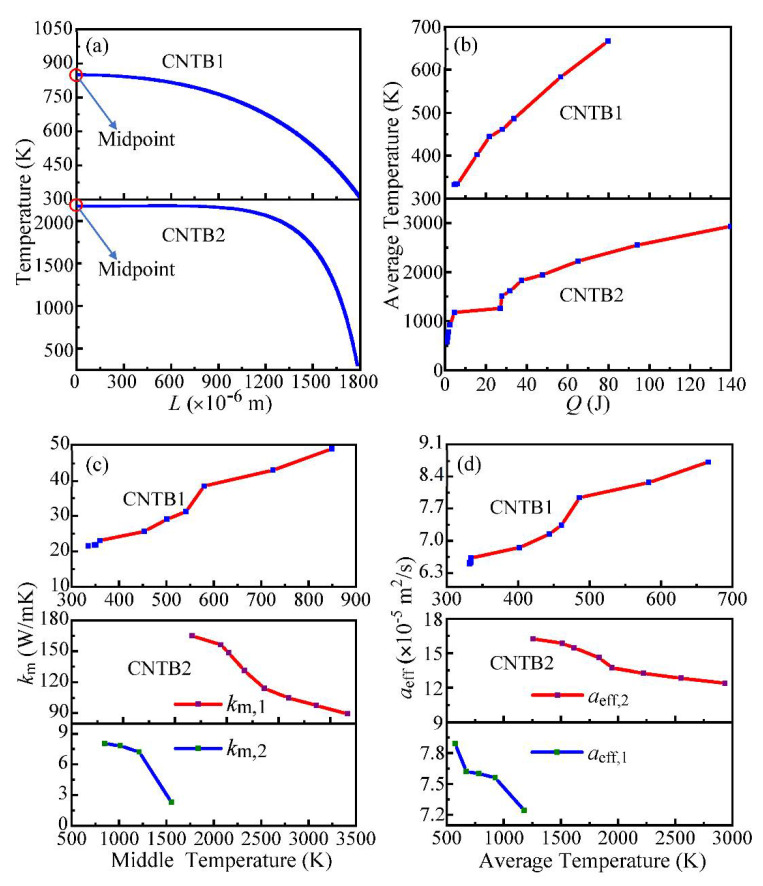
(**a**) Temperature distribution along the length of CNTB. The x-coordinate zero is the midpoint of the sample. (**b**) The average temperature of CNTB under different heating conditions. (**c**) Thermal conductivity of the midpoint varies with the annealing temperature of the midpoint. *k*_m,1_ and *k*_m,2_ correspond to the low thermal conductivity before switch-on and the high thermal conductivity after switch-on of CNTB2. (**d**) The thermal diffusivity varies with the average annealing temperature. *α*_eff,1_ and *α*_eff,2_ also correspond to the states before and after switch-on of CNTB2.

**Figure 8 nanomaterials-12-00083-f008:**
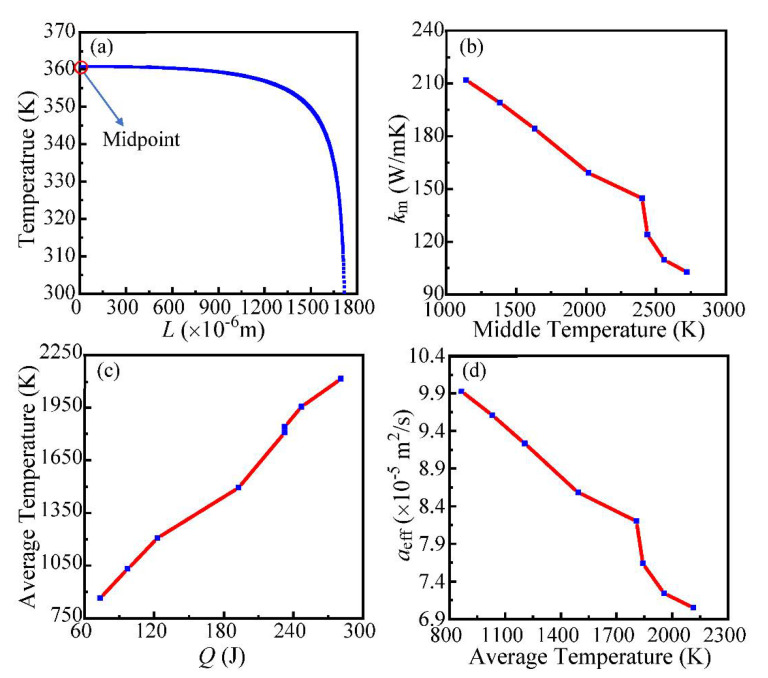
(**a**) Temperature distribution along the length of CNTF. The x-coordinate zero is the midpoint of the sample. (**b**) Thermal conductivity of the midpoint varies with the annealing temperature of the midpoint. (**c**) The average temperature of CNTF under different heating conditions after switch-on. (**d**) The thermal diffusivity varies with the average annealing temperature.

**Figure 9 nanomaterials-12-00083-f009:**
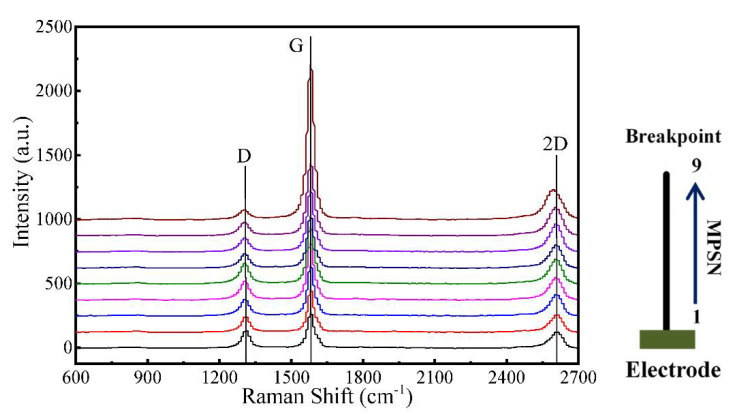
The Raman spectrum with measuring point positions 1–9 of the burned CNTB sample. Point 1 is the furthest from the breakpoint, and 9 is the closest.

**Figure 10 nanomaterials-12-00083-f010:**
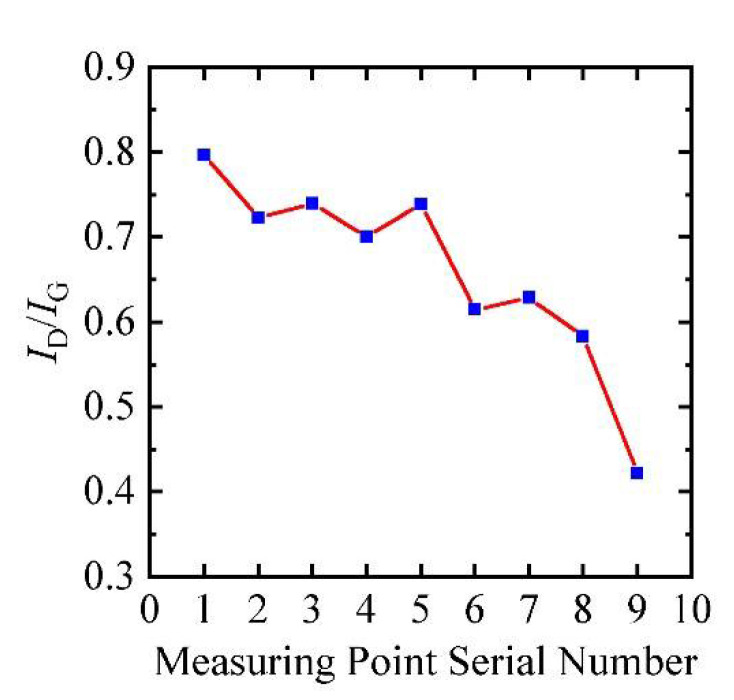
The *I*_D_/*I*_G_ of CNTB varies with the position of measuring points 1–9.

**Table 1 nanomaterials-12-00083-t001:** Partial experimental results of CNTB1 during current annealing.

Sample	*I*_a_ (×10^−3^ A)	*α*_eff_ (×10^−5^ m^2^/s)
CNTB1	0	6.52
	90	6.53
	120	6.85
	150	7.34
	180	8.27
	200	8.71

**Table 2 nanomaterials-12-00083-t002:** Partial experimental results of CNTB2 and CNTF during current annealing.

Sample	*I*_a_ ×10^−^^3^ A	Low-Level Thermal Diffusivity after Annealing $\left(\alpha_{\mathrm{eff},1}\right)$	High-Level Thermal Diffusivity after Annealing $\left(\alpha_{\mathrm{eff},2}\right)$	Low-Level Thermal Diffusivity before Annealing $\left(\alpha_{\mathrm{eff},01}\right)$	High-Level Thermal Diffusivity before Annealing $\left(\alpha_{\mathrm{eff},02}\right)$
		×10^−5^ m^2^/s		×10^−5^ m^2^/s	
CNTB2	40	7.89		8.09	
	100	6.79			
	140	5.41			
	150	13.02	13.72		
	160	6.59	32.01		
	165	8.12	24.02		
	170		16.25		
	210		13.72		
	300		12.38		
CNTF	300	4.27	18.90	3.45	21.19
	350	5.10	19.63		
	400	4.94	18.95		
	450		9.93		
	675		8.20		
	720		7.05		

## Data Availability

The data presented in this study are available on request from the corresponding author.

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
