# Peer review of "Effects of Current Annealing on Thermal Conductivity of Carbon Nanotubes"

_nanomaterials, 2021, doi:10.3390/nano12010083_

Round 1
Reviewer 1 Report
Effects of Current Annealing on Thermal Conductivity of Carbon Nanotubes
Manuscript ID: nanomaterials-1512911
The authors have conducted a study aiming at enhancing thermal conductivity properties of carbon nanotube films by using a current annealing treatment method. The thermal investigation part is rich and well described. However, the main weakness of the work concerns the lack of material characterization to support the given claims. Several improvements are required and additional investigations are needed to strengthen the study and the author’s claims. The authors propose several mechanisms to explain the thermal measurements. Especially structural modifications such graphitic order modification, reduction in defect density, impurity moving/elimination, graphitization degree… and these latter are not enough supported by systematic characterization of MWCNTs such as Raman spectroscopy, TEM or any other relevant technique. And the abstract needs to be rewritten since no sample name has to be given in the abstract.
Reviewer 2 Report
The manuscript is appropriate to be published in the Nanomaterials journal; however, some issues must be answered supported in experimental evidence. For example, there are no SEM images showing samples' anisotropic characteristics. Below are the main issues that need to be solved.
The following paragraph is unnecessary because they report improvements in thermal and electrical properties in graphene, not in carbon nanotubes or fibers.
"Xin et al.[11] identified that highly arranged defect-free graphene paper treated with high temperature annealing has excellent thermal and electrical properties, of which thermal conductivity is up to 1434
W/ m K , which is a 400% improvement compared to the GPs before annealing. Song et al.[12] conducted thermal annealing treatment on graphene oxide (GO) films and found that the oxygen content of GO decreases with the increase of temperature, reaching the peak of heat transfer at 1200 ℃ with the thermal conductivity of 1043.5 W/ m K."
Please explain what the capricious process means in this scenario. Capricious is a subjective word.
In figure 1, please clarify that the showed CNBT is before the TET experiments.
Please explain why the unusual phenomenon (sudden jump) found in CBTB2 proves that the samples are anisotropic. What is the origin of such anisotropy? Please, explain the jump from a morphological point of view.
Please include SEM images after the TET experiments.
In conclusion: CNTB1 is repeated: "The difference between CNTB1 and
CNTB1 is caused by the anisotropy of CNTB"
If the only difference between CBTN1 and CBNT2 is the direction of preparation, why the impurities are relevant for the anisotropy behavior. Please explain in detail.
What happens when the direction of preparation is 45 degrees?
Reviewer 3 Report
The authors have observed interesting variations of thermal properties of multi-walled carbon nanotube films and carbon fiber films with Joule heating of the samples. They attempt to explain the underlying physics, which is of course very complex. The main weak point of the paper is its partial lack of clarity. This must be improved. The reproducibility of the data is an other important issue not discussed in the paper.
1) Important points to clarify
It is not clear what sample was used in the TET experiments. Was it a rectangle cut from the 10x10 cm film along one side for CNT1 and along a perpendicular direction for CNT2? Same question for CNTF.
The section "Experimental procedure" in P. 5 should be improved. It is not clear if the annealing current is increased in a stair-wise manner or if the annealing consists of a succession of current pulses of higher and higher intensities? In either case, the time duration of the steps or pulses should be given.
The discussion in Section 3.1.3 is difficult to follow. In particular, what is the meaning of the sentence "During the TET measurements the independent structure of the sample will no longer be reversed". How do the authors make a distinction between "pure" carbon nanotubes (P-CNT) and impurity-embedded carbon nanotubes (I-CNT)? Why and what kind of impurity do they have in mind? Why should the sample CNT2 be a mixing of both and not sample CNT1?
In the long discussion of phonon scattering starting P. 14, is reference 20 really relevant for multi-walled carbon nanotubes and carbon fibers, where both structural and atomic disorders are present?
In P. 16, the authors write "With the enhancement of annealing, the sample becomes orderly, the disorder of internal structure decreases, and the defect density also". Do they have recorded Raman spectra at different intensities of the annealing current to support that statement? Do they have microscopic images of the structure of the samples after different annealings?
What about reproducibility of the observations? For instance, does the low to high α switch-on annealing current vary from one CNTB2 sample to another?
2) Some minor things
Please use the widely accepted acronym MWCNT for multi-walled carbon nanotubes instead of mCNT.
The text in Sect. 3.1.1. refers to Table 1 and describes the variations of αreal. By contrast, Table 1 gives values of αeff. Please, check which one is under discussion.
P. 3: the authors write "self-designed sample holder". Do they mean "home-made sample holder" (meaning designed by them)?
P. 4: please define the parameters of eq. (1), more specifically q0, k, η.
P. 4, a few line after eq. (2): the author write "eventual voltage". Do they mean the "final voltage"?
P. 4, near the bottom: please correct "sociated with" into "associates with".
In Fig. 3, the diffusivity is designated with the letter "a" whereas the symbol "α" is used in the text of P. 7. Same remark for Fig. 5 and the discussion in P. 8. Please be coherent.
Caption of Fig. 11 and reference to it in P. 16: the authors write "optical phonon free path". Do they mean "optical phonon mean free path"?
In the conclusion, please change the second "CNTB1" into "CNTB2" in the sentence "The difference between CNTB1 and CNTB1"
Round 2
Reviewer 1 Report
The authors have responded to my comment and have made the required modifications to the revised manuscript
Reviewer 2 Report
The authors answered the questions positively and included the suggestions in the manuscript. The research work is ready to be published on Nanomaterials Journal.
Reviewer 3 Report
The manuscript has thoroughly been revised by the authors. The present version is acceptable fot publication in Nanomaterials.